# Outcome Through the Years of Left-Ventricular Assist Devices Therapy for End-Stage Heart Failure: A Review

**DOI:** 10.3390/jcm13216622

**Published:** 2024-11-04

**Authors:** Ilaria Tropea, Giovanni Domenico Cresce, Valerio Sanesi, Loris Salvador, Daniele Zoni

**Affiliations:** Cardiovascular Department, S. Bortolo Hospital, 36100 Vicenza, Italy; giovannidomenico.cresce@aulss8.veneto.it (G.D.C.); valerio.sanesi@aulss8.veneto.it (V.S.); loris.salvador@aulss8.veneto.it (L.S.)

**Keywords:** heart failure, left-ventricular assist devices, bleeding, infection, right ventricle dysfunction

## Abstract

Heart transplantation remains the gold standard surgical treatment for advanced heart failure. Over time, medical therapies have achieved remarkable outcomes in terms of survival and quality of life, yet their results may be insufficient, even when maximized. The limited availability of organ donors and the selective criteria for heart transplant eligibility have led to a significant rise in the utilization of long-term mechanical circulatory support, including left ventricular assist devices. Patients receiving LVADs often present with multiple comorbidities, constituting a highly vulnerable population. Individuals living with LVADs may experience various long-term complications, such as bleeding, driveline infections, neurological events, and right ventricular dysfunction. Fortunately, the development of increasingly biocompatible LVAD devices in recent years has resulted in a notable reduction in these complications. This review aims to summarize the principal complications encountered by patients with LVADs throughout their treatment and the associated daily management strategies.

## 1. Introduction

Heart failure represents a complex clinical syndrome defined by a collection of signs and symptoms stemming from structural or functional impairment of the heart and its ability to effectively pump blood [1]. The first-line treatment is represented by medical therapy; however, for patients whose medical therapy has reached its maximum potential without producing tangible clinical improvements, enrollment in an advanced treatment modality may be warranted.

Heart transplantation remains the surgical benchmark for managing end-stage heart failure, boasting a one-year survival rate of approximately 90% and an overall mean survival of 12.5 years [2]. The lack of donors and the adverse effects associated with prolonged immunosuppressive medical treatment have motivated the exploration of alternative therapeutic approaches for patients who continue to experience disease progression despite optimized medical management [3]. The management of individuals ineligible for cardiac transplantation was transformed by the advent of left-ventricular assist devices. These devices are capable of pumping blood into the aorta, bypassing the malfunctioning left ventricle through the utilization of a pump powered by a cable linked to an external energy supply [2].

LVADs have become a valuable option for patients with end-stage heart failure, either as a bridge to cardiac transplantation or as a destination therapy; the survival of patients treated with LVADs has significantly improved over the years as a result of technological advances and improved patient management strategies [4].

First-generation LVADs produced a pulsatile flow, aiming to mimic the function and physiological blood flow of the heart. However, the sizeable dimensions of these devices, the high incidence of adverse events, and the rapid deterioration of materials necessitated a series of modifications to the system, ultimately leading to the development of second-generation LVADs [5]. These devices featured a single axial pump that generated a continuous flow. While outcomes saw significant improvement, the rates of pump thrombosis and adverse events remained high. The introduction of magnetic levitation technology led to the development of third-generation left-ventricular assist devices currently in clinical use [5].

Through the years, a series of trials have been conducted to examine the short-term and long-term complications associated with left-ventricular assist device implantation, as well as to evaluate the outcomes of patients who had undergone this procedure.

This review aims to synthesize the existing literature on the long-term adverse events experienced by LVAD patients and their management in everyday life.

## 2. End-Stage Heart Failure: When Medical Therapy Is Not Enough

Patients diagnosed with heart failure initially receive medical treatment, with medication dosages typically increased to the maximum tolerable level. Additionally, these individuals often undergo palliative procedures aimed at alleviating symptoms and enhancing their quality of daily life, such as electrophysiological resynchronization therapies and percutaneous interventions for functional valvular disorders [1,3]. Symptomatic patients, despite maximal medical therapy, may be assessed for suitability for surgical intervention for end-stage heart failure, with heart transplantation considered the ideal treatment approach. Criteria for inclusion on the heart transplant waitlist are highly selective, and a range of factors, such as advanced age (over 70 years), active infections, pulmonary hypertension, cerebrovascular and peripheral vascular disease, and obesity (BMI exceeding 35 kg/m^2^), are typically regarded as contraindications [6]. The lack of donors is the main limit of heart transplants, further reducing the number of patients eligible for this treatment; this series of limitations have prompted the development of alternative strategies over the years for patients who, despite medical management, experience disease progression, such as mechanical circulatory support. Mechanical circulatory support has been shown to improve the symptoms and survival of individuals afflicted with end-stage heart failure [7]. The use of long-term mechanical circulatory support is guided by the INTERMACS classification system and can be applied to various clinical scenarios: bridge to candidacy, where it is utilized for patients currently ineligible, with the aim of improving their hemodynamic status; bridge to transplant, for patients without an immediately available donor, allowing them to await a suitable transplant; bridge to recovery, for patients experiencing reversible cardiogenic shock; destination therapy, for those not eligible for heart transplantation due to the established listing criteria [7] [Table 1].

## 3. LVAD: A Machine to Supplement the Heart

In 1966, DeBakey conducted the inaugural successful implantation of a left-ventricular assist device, utilizing a paracorporeal pneumatically powered pump as a bridge to recovery in a patient experiencing post-cardiotomy syndrome. The patient subsequently recovered, and the device was removed 10 days later [8]. In 1978, Norman implanted the first LVAD as a bridge to transplant; it was an intracorporeal pump located in the patient’s abdomen [9]. The first implant as destination therapy was performed in 2002, thanks to the REMATCH Trial that compared outcomes between long-term use of LVADs and medical therapy for end-stage heart failure [10]. The REMATCH study revealed a substantial enhancement in patient survival and quality of life, establishing left-ventricular assist devices as a viable alternative for individuals ineligible for heart transplantation [11].

First-generation LVADs were devices that produced a pulsatile flow, whose function was meant to mimic the normal cardiovascular physiology and particularly the pump role of the heart. This category included LVAS (left-ventricular assist system) like HeartMate XVE® (Thoratec Corporation, Pleasanton, CA, USA) and Novacor® (Baxter Healthcare Corporation, Novacor Division, Oakland, CA, USA) [5].

Despite first-generation LVADs representing a revolution and innovation for the surgical therapy of end-stage heart failure, these devices were associated with a range of limitations and adverse effects. Their bulky size made them challenging to transport, and their batteries had a limited lifespan (5 h). Additionally, the materials used in these devices were prone to deterioration and pump malfunctions. Furthermore, the risks of hemorrhage, thrombosis, and infection were elevated. To address these shortcomings, second-generation LVADs were subsequently developed [12].

Second-generation LVADs included models like HeartMate II® (Thoratec Corporation, Pleasanton, CA, USA) [Figure 1] and Jarvik 2000^®^ (Jarvik Heart, Inc.; New York, NY, USA); these LVADs were constituted of a single axial rotating pump generating a continuous flow [12]. The new design and the smaller dimensions extended LVAD implantation to a wider population; the inner single rotor reduced the number of components undergoing wear and tear [13], but this new conformation required a systemic anticoagulation therapy, not mandatory in first-generation LVADs since the direct contact between the blood and the rotor augmented the risk of pump thrombosis [5].

Second-generation LVADs brought a significant increase in survival compared to first-generation ones; however, the morbidity and mortality of patients implanted with LVADs were not satisfying when compared to patients who had undergone heart transplantation.

Third-generation LVADs are characterized by a further reduction in dimension, major biocompatibility, and a lower risk of pump thrombosis; they are made of a centrifugal pump generating a continuous flow, and they include HeartMate 3® (Thoratec Corporation, Pleasanton, CA, USA) [Figure 2] and the HeartWare HVAD system® (Heart-Ware International, Inc., Framingham, MA, USA), which use, respectively, a full and hybrid magnetic levitation system [13]. In recent years, an increased incidence of cerebrovascular events has been observed in patients receiving HeartWare left-ventricular assist devices. Consequently, on 3 June 2021, the FDA and Medtronic issued a statement to discontinue the distribution and sale of the Medtronic HeartWare System [14].

## 4. Evolution in Mechanical Circulatory Support: From Axial to Centrifugal Flow Pumps

In September 1994, the FDA approved the use of left-ventricular assist devices as a bridge to transplant. In the subsequent years, multiple studies demonstrated that patients implanted with LVADs exhibited improved hemodynamics, better organ perfusion, enhanced exercise tolerance, and overall increased survival compared to those receiving medical therapy alone. The REMATCH Trial further expanded the indication for LVAD implantation as destination therapy, comparing outcomes between LVADs and optimal medical management. This randomized trial enrolled 129 patients who were assigned to either HeartMate VE implantation or maximal medical therapy. The 1-year survival rate was 52% in the LVAD group, compared to 23% in the medical therapy cohort. However, the 2-year survival rates declined to 23% and 8% in the LVAD and medical therapy groups, respectively. The causes of death also differed between the two groups, with the medical therapy patients predominantly succumbing to left-ventricular dysfunction, while the LVAD recipients experienced complications such as sepsis, device malfunction, cerebrovascular events, and pulmonary embolism, with only one case of left-ventricular dysfunction [10]. While pulsatile left-ventricular assist devices demonstrated improvements in patient survival, they were also associated with a significant range of complications, including infections, cerebrovascular events, and pump failure. Consequently, a series of modifications were required, with the primary change being the transition from pulsatile to continuous-flow LVAD technology [15]. The second-generation LVAD technology employed a single axial rotating pump, but this design entailed a direct blood–pump interaction, significantly increasing the risk of pump thrombosis. However, initial trials of second-generation LVADs demonstrated improved survival rates compared to the first-generation devices. As reported by Slaughter and colleagues, patients with continuous-flow LVADs exhibited a higher probability of surviving and being free from cerebrovascular events and device malfunctions at the 2-year mark than those with pulsatile-flow LVADs [16]. However, from 2011 an increase in the incidence of pump thrombosis was recorded in patients with HeartMate II in three centers in the U.S.: Starling et al. reported an increase in pump thrombosis from 2.2% to 8.4%, with a consequent worsening in terms of survival at 180 days from the diagnosis, and the mortality was 35.6% in patients with pump thrombosis vs. 16.8% in patients without pump thrombosis [17]. In addition, patients implanted with a second-generation LVAD required oral anticoagulation therapy, and this led to a major risk of gastro-intestinal bleeding; this condition was both caused by the oral anticoagulation therapy and the vascular physiology of patients with a continuous blood flow (angiodysplasias and acquired Von Willebrand disease) [18]. To obviate these limits, the main innovations of third-generation LVADs included no direct contact between the blood and the pump, a major biocompatibility of the materials, and an oscillation in pump speed that conferred a pulsatile-like movement to the blood flow. HeartWare was the first third-generation LVAD introduced into clinical practice, and it was launched by the ADVANCE Trial: a population of 140 patients eligible for a heart transplant was selected to be implanted with HeartWare as a bridge to transplant. The 180-day outcomes were compared to a control group with HeartMate II devices from the INTERMACS Registry. The ADVANCE Trial established the non-inferiority of third-generation HeartWare devices compared to second-generation left-ventricular assist devices in terms of survival [19]. The ENDURANCE Trial, published in 2017, aimed to compare the safety and efficacy of centrifugal flow LVADs versus axial flow LVADs. The study enrolled 446 patients, with 297 assigned to the centrifugal flow LVAD and 158 to the axial flow control device. The primary endpoint was 2-year survival free from stroke or device malfunction. The results showed that the centrifugal flow LVAD was non-inferior to the axial flow device, with 55% and 57.4% of patients, respectively, achieving the primary endpoint. There were no significant differences in mortality or stroke rates between the groups, but the control group had a higher incidence of device malfunction. Adverse events such as bleeding, arrhythmias, and local infections were similar. However, the centrifugal flow LVAD group experienced a higher rate of stroke, particularly in the first six months, and increased stroke risk was associated with mean arterial pressure over 90 mmHg. Conversely, the control group had a higher incidence of pump thrombosis and device malfunction [20]. The unexpected risk of stroke of third-generation LVADs was investigated successively with the ENDURANCE supplemental trial, whose aim was to determine prospectively the effectiveness of an antihypertensive therapeutic strategy to reduce the risk of cerebrovascular events in patients with LVADs. The data obtained from the two studies permitted the approval, in 2017, by the FDA of HVAD as a destination therapy. The ENDURANCE supplemental trial randomized 465 patients to study devices (HeartWare HVAD) or to control devices (HeartMate II); the primary endpoint was to evaluate the 12-month incidence of stroke, while the secondary endpoints included freedom from death, stroke, heart transplants, and a malfunctioning device needing replacement. At the end of the trial, the primary endpoint was not achieved since the incidence of stroke was 14.7% in patients with HeartWare versus 12.1% with HeartMate II (*p* = 0.14), while the secondary endpoint showed the superiority of the study device (76.1%) versus the control device (66.9%) (*p* = 0.04). The supplemental trial demonstrated a reduction in the incidence of stroke in patients with HVAD compared to the first trial, thanks to the pressure control protocol [21]. Third-generation LVAD HeartMate 3 was released to contrast the previous generation thrombotic phenomena, and it constituted a centrifugal flow pump with a fully magnetic levitation mechanism. It was launched by the MOMENTUM 3 Trial (Multicenter Study of MagLev Technology in Patients Undergoing Mechanical Circulatory Support Therapy with HeartMate 3), whose aim was to compare the outcome of patients implanted with HeartMate II or HeartMate 3. This trial involved 294 patients randomized to HeartMate II or HeartMate 3; of 294 patients, 152 were assigned to the centrifugal flow pump group, while 142 patients were assigned to the axial flow pump group; five patients (one in the study group and four in the control group) did not receive any device. The primary endpoint was survival and free from a disabling stroke or device malfunction requiring reoperation to remove the device at 6 months from the implantation. According to the MOMENTUM 3 Trial, the primary endpoint was reached by 131 patients (86.2%) of the study device and 109 (76.8%) of the control one. The two groups did not show a significative difference in terms of incidence of death and disabling stroke; however, reoperation for device malfunctioning was more frequent in the control device rather than in the study device, with a percentage, respectively, of 7.7% versus 0.7%. Of the 141 patients with a centrifugal flow pump, no pump thrombosis was recorded, while 14 patients (10.1%) of the axial flow group had a thrombotic event (*p* < 0.001). The most frequent causes of death were stroke, right ventricle dysfunction, and infection, indifferently in the two groups. In conclusion, the MOMENTUM 3 Trial demonstrated how patients implanted with centrifugal flow pumps had a better outcome at 6 months [22]. The MOMENTUM 3 Trial progressively expanded both the patient population and the duration of the follow-up, gathering data from 1028 participants. Of these, 516 patients were randomly assigned to a centrifugal flow device, while 512 were allocated to an axial flow device. Ultimately, 515 patients received the HeartMate 3, 505 received the HeartMate II, and 6 did not receive any device. The primary endpoint was 2 years of survival and free from a disabling stroke and pump replacement due to device malfunction. The secondary endpoint was device replacement at 2 years post-implantation. The analysis of the primary endpoint demonstrated how 397 (76.9%) patients in the centrifugal flow group had a 2-year survival rate and were free from a disabling stroke or device replacement due to malfunction versus 332 (64.8%) patients in the axial flow group (*p* < 0.001); 57 patients (11.3%) of the axial flow group required a device replacement compared to 12 patients (2.3%) in the other group (*p* < 0.001). In relation to the secondary endpoint, pump replacements were less frequent in patients with HeartMate 3 (12 [2.3%]) than in patients with HeartMate II (57 [11.7%]; overall, patients with a centrifugal pump flow had a lower incidence of adverse events like bleeding, stroke, and gastro-intestinal hemorrhage. In the end, the final report of the MOMENTUM 3 Trial demonstrated the superiority of centrifugal flow pumps compared to axial flow ones [23].

## 5. Outcome of Patients with an LVAD

The number of patients receiving left-ventricular assist devices has been steadily rising over the years. While short-term survival rates on LVAD support resemble those seen after heart transplantation, long-term outcomes still tend to favor heart transplantation. There have been a limited number of studies directly comparing the safety and effectiveness of mechanical circulatory support devices versus heart transplantation [24]. As reported by Theochary et al., 1-year survival did not differ significantly between heart transplant and LVAD groups, either as a bridge to transplant or destination therapy [25]. Comparative long-term outcomes indicate that heart transplantation yields superior results, with a median survival of 12.5 years [2] compared to 7.1 years for patients with continuous-flow left-ventricular assist devices, excluding the HeartMate3 model. Furthermore, patients with LVADs experience a higher rate of hospital readmissions following device implantation, in contrast to those who undergo heart transplantation [24].

The advent of HeartMate 3 improved the results of patients with LVADs, as reported by the ELEVATE Registry in a 5-year follow-up: patients with HM 3 had a 5-year survival rate of 63.3% in patients affected by end-stage heart failure; stroke represented one of the most important complications and the major cause of morbidity and mortality [26].

In addition, a progressively aging heart failure population, coupled with advancements in mechanical circulatory support technology, has led to a rise in the number of left-ventricular assist device implantations among older adults (>75 years) with end-stage heart failure. This trend reflects the growing demand for effective treatment options for this vulnerable patient population, as well as the continued improvements in LVAD design and functionality that have expanded the suitability of this therapy for an increasingly diverse range of patients. Caraballo et al. analyzed a total of 20,939 patients undergone the implant of LVAD, 4.9% of which were ≥75 years of age, reporting that this subgroup had an increased mortality rate after LVAD implantation. Older patients with left-ventricular assist devices experienced a higher frequency of gastro-intestinal bleeding but a lower incidence of device thrombosis compared to their younger counterparts. Furthermore, while 84.5% of individuals under 55 years old were able to be discharged home following LVAD implantation, this was only true for 46.8% of adults aged 75 and above. Key predictors of outcomes in the oldest patient group included the use of a right ventricular assist device, serum albumin levels, and the 6 min walking test [27]. On the other hand, a study conducted by Emerson et al. examined a population of patients receiving LVAD between 2010 and 2020; the cohort included 68.9% (*n* = 16,808) patients under 65 years, 26.3% (*n* = 6418) patients aged between 65 and 75 years, and 4.8% (*n* = 1182) patients older than 75 years, who were mostly male (*n* = 19,119, 78%) and on destination therapy (*n* = 12,425, 51%). Mortality rates were found to be higher in older age groups, with 34% for patients under 65 years, 54% for those aged 65–75 years, and 66% for those over 75 years. However, mortality improved over the course of the study for the oldest age group. Newer-generation left-ventricular assist devices were associated with reduced late mortality. As the patient’s age increased, the incidence of stroke, device malfunction or thrombosis, and rehospitalization decreased. Additionally, the median 6 min walk distance and quality of life measures improved after LVAD implantation across all age groups [28].

Annually, the Society of Thoracic Surgeons Interagency Registry for Mechanically Assisted Circulatory Support publishes a report detailing the outcomes of LVAD recipients.

The 14th annual INTERMACS registry report analyzed data from 27,493 patients who underwent continuous-flow LVAD implantation during the decade spanning 2013 to 2022. The shift to the mostly exclusive use of fully magnetic levitated devices (Mag-Lev) led to a comparison of the outcomes between Mag-Lev LVADs and contemporary (from 2018 to 2022) and historical non-Mag-Lev ones (from 2013 to 2017). As reported by the INTERMACS registry, the 1- and 5-year survival rates were higher in patients with Mag-Lev devices (respectively 86% and 64%) compared to patients with contemporary and historical non-Mag-Lev ones (79% and 44% at 1 year, and 81% and 44% at 5 years) (*p* < 0.0001). In terms of adverse event [Figure 3], 5-year freedom from stroke (87% versus 67%, *p* < 0.0001), gastro-intestinal bleeding (72% versus 70%, *p* < 0.0001), and device malfunction (83 vs. 54%, *p* < 0.0001) were definitely higher in patients with Mag-Lev devices than those with contemporary non-Mag-Lev ones; on the other hand, device-related infections were lower in the non-Mag-Lev group than in the Mag-Lev one (respectively 61% versus 64%, *p* = 93). The causes of death in the Mag-Lev cohort were a withdrawal of support (19%), multiorgan failure (16%), and heart failure (14.2%). A conspicuous reduction in the percentage of deaths related to neurological dysfunction emerged in the Mag-Lev group (9.2%) versus the contemporary (12.7%) and historical (17.6%) non-Mag-Lev ones [29].

### 5.1. Cerebrovascular Events

The incidence of cerebrovascular events in patients supported with LVADs goes from 15% to 29% per year, and a series of risk factors and mechanisms are implied [30]. The contact between the blood and the device, the suction of thrombus from a dilated left ventricle, and the use of anticoagulant and antiplatelet therapies represent the main risk factors [31]. In addition, common cardiovascular risk factors (atrial fibrillation, previous stroke, diabetes, and dyslipidemia) sum up the presence of the device; risk factors for ischemic stroke are inadequate antiplatelet therapy, pump thrombosis, and hypertension (MAP over 90 mmHg), while risk factors for hemorrhagic stroke are elevated INR, hypertension, acquired Von Willebrand disease, and ischemic stroke [21,31].

A good pressure control protocol and an adequate anticoagulant therapy lower the incidence of both ischemic and hemorrhagic strokes [21]; as reported by Inamullah et al., of 247 patients, 12.1% had a stroke, 63% of which were ischemic; INR was not therapeutic in 47.4% of ischemic strokes, and it was too high in 18.2% of hemorrhagic ones [32].

According to Kadakkal et al., the female sex constitutes a risk factor for stroke, especially hemorrhagic, as reported by several studies [33]. Kirklin et al. analyzed a total of 9489 patients from the INTERMACS registry implanted with axial and hybrid (magnetic and hydrodynamic) levitated centrifugal flow devices between 2014 and 2017. During the follow-up, 1515 (16%) patients experienced one or more cerebrovascular events, and the risk of stroke was 4% during the first month, 6% in the first six months, and 14% in the first year. In addition, disabling stroke was associated with a worse prognosis compared with a stroke that led to mild or no disability [34].

Despite the advancements in the design and biocompatibility of third-generation left-ventricular assist devices, stroke remains one of the primary adverse events, partially attributable to non-modifiable patient-related risk factors [35].

### 5.2. Gastro-Intestinal Bleeding

Gastro-intestinal bleeding (GIB) is reported in one-third of patients supported by LVADs [31], causing frequent hospitalizations and impairing the quality of life. The pathophysiology of gastro-intestinal bleeding is multifactorial: the loss of pulsatile flow through the vessels and the presence of a continuous flow leads to an acquired Von Willebrand disease, similar to that observed in aortic stenosis, which increases the bleeding risk. Furthermore, the continuous flow causes relative hypoperfusion of the gut, prompting the development of angiodysplasias, which are prone to bleeding. Additionally, patients with LVADs typically receive antiplatelet and anticoagulant therapies, which further contribute to the risk of gastro-intestinal bleeding. This complex interplay of factors underlies the gastro-intestinal bleeding experienced by individuals undergoing LVAD therapy [36].

Hammer et al. collected data from 6425 patients enrolled in the INTERMACS Registry from 2017 to 2020 to evaluate the impact of gastro-intestinal bleedings in patients supported by third-generation LVADs: of 6425 patients, 1010 (15.7%) suffered from GIB, and while post-LVAD-implantation GIB was related to mortality, there was no correlation between the number of GIB episodes and death [37].

Hallet et al. reported the incidence of GIB in a population of continuous-flow LVADs implanted between 2006 and 2014: of 10470 patients, 2717 (26%) had a GIB, and 1316 (12.6%) experienced more than one episode of GIB; 49.2% of GIB required hospitalization, 88.7% of which needed transfusion, and 0.7% conducted the patient to death. Identified risk factors were the female sex, older age, low preoperative hemoglobin, and low INR after implantation [38].

To prevent and reduce the risk of GIB, a series of medical treatments have been suggested, like octreotide, angiotensin-converting enzyme inhibitors, thalidomide, and digoxin [34].

### 5.3. Infections in Patients with LVADs

The classification of LVAD-related infections was defined by The Infectious Diseases Council of the International Society of Heart and Lung Transplant (ISHLT) and recognizes three different types: VAD-specific, VAD-related, and non-VAD infections [39]. VAD-specific infections involve parts of the device: the driveline, the driveline tunnel, the pump, the inflow, and the outflow cannula; they may be transferred at the moment of the implant or later through the driveline or other infective foci; VAD-related infections are related to the implant: endocarditis, mediastinitis, and sepsis; non-VAD infections refer typically to pneumonia or urinary tract infection, i.e., parts not related to the presence of the device [40]. Most of the infections in patients supported by LVADs derive from the driveline, and risk factors are represented by older age, obesity, diabetes, chronic kidney disease, and long-time LVAD [41].

As reported by Kusne et al., LVAD infections cause an augmented rate of rehospitalization and morbidity and mortality in patients who do not undergo a heart transplant; in fact, a superficial driveline infection (DLI) may become a deep tissue infection, developing sepsis [41,42].

To reduce the risk of VAD infections and particularly DLI, a main role is played by prevention and early treatment [43].

## 6. Conclusions: Looking at the Future, What Is Next?

Left-ventricular assist devices have emerged as a viable alternative for patients with end-stage heart failure who are ineligible for heart transplantation. However, these devices are still associated with frequent adverse events, often linked to the device itself.

Significant limitations include mandatory anticoagulant and antiplatelet therapy, the presence of the driveline, and limited battery life, all of which can impact the daily lives of LVAD patients and diminish their quality of life.

While the introduction of the HeartMate 3 has allowed for lower international normalized ratios, an ideal LVAD would not require anticoagulant therapy and would not have an exit site, being perfectly biocompatible and truly integrated with the patient’s own heart. This obstacle may be overcome thanks to soft robotics, developing devices compressing directly the heart or the aorta and not having direct contact between the machine and the blood. On the other hand, biocompatibility may be improved by hybrid technologies that eliminate the blood-device surface area, seeding endothelial cells on the device surface [44].

Fortunately, the field of engineering continues to advance, and we will probably see the development of novel devices that offer more physiological and synchronized pulsatile flows that are way more biocompatible and forgettable (without any external component, like the driveline) [45].

## Figures and Tables

**Figure 1 jcm-13-06622-f001:**
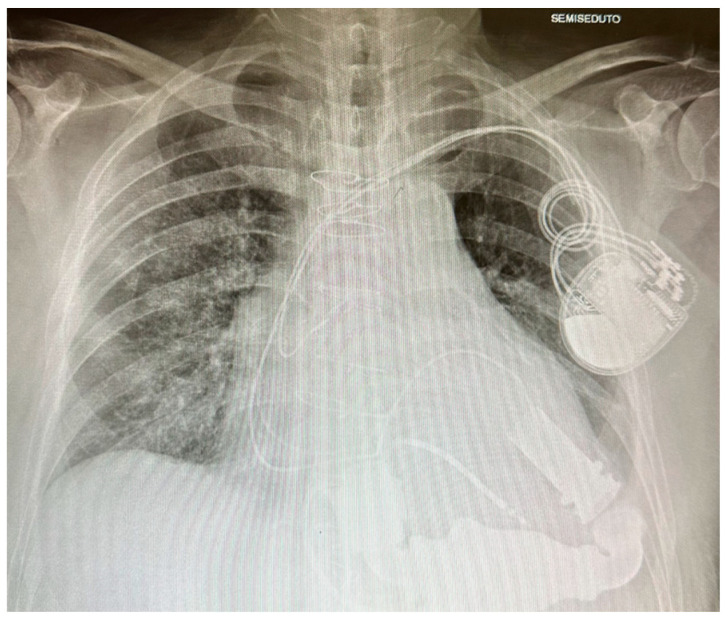
HeartMate II.

**Figure 2 jcm-13-06622-f002:**
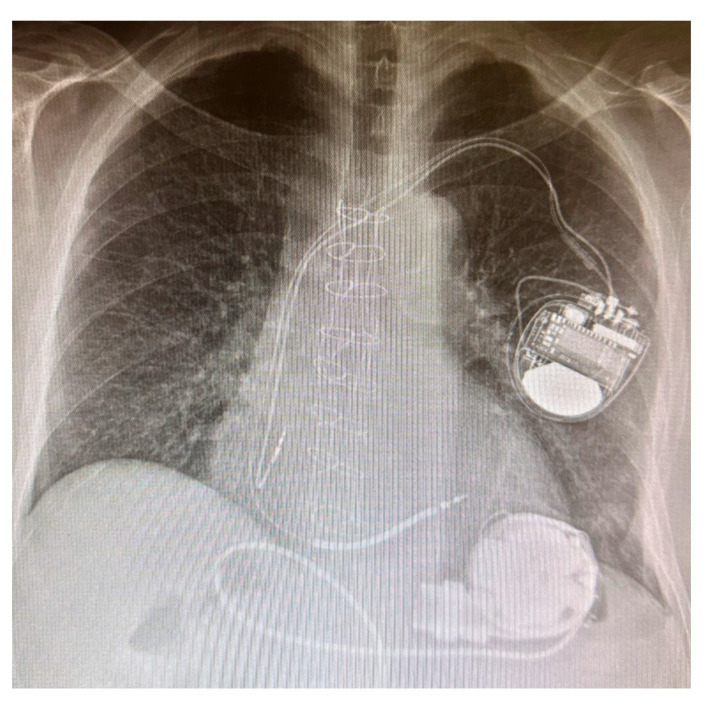
HeartMate 3.

**Figure 3 jcm-13-06622-f003:**
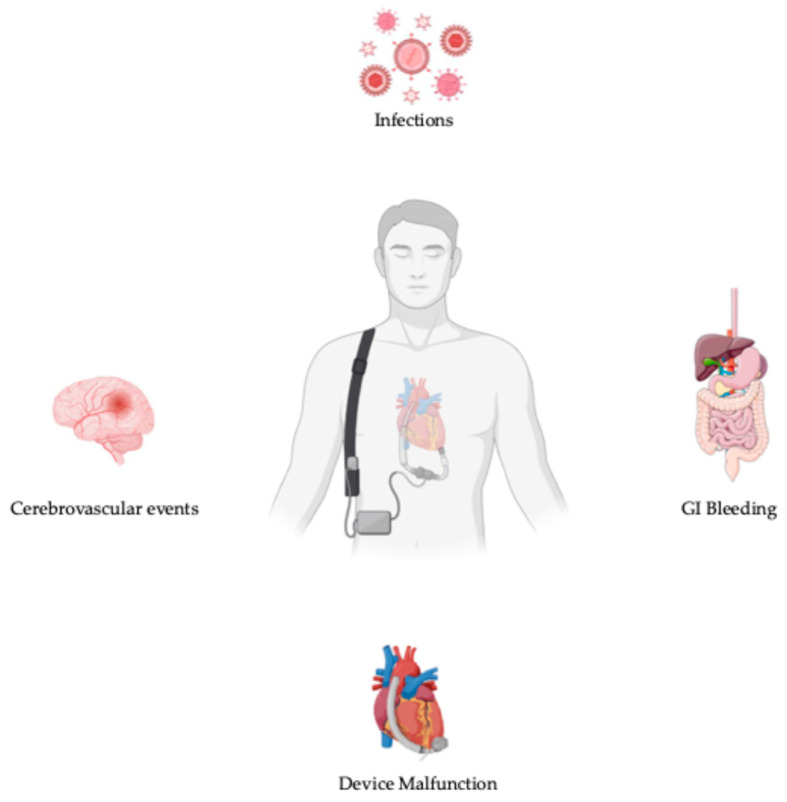
Adverse events in patients with LVADs.

**Table 1 jcm-13-06622-t001:** Indications for MCS.

BTD (Bridge to Decision) BTB (Bridge to Bridge)	Short-Term MCS in patients with drug-refractory cardiogenic shock at immediate risk of death to sustain life until a clinical evaluation is completed
BTC (Bridge to Candidacy)	Long-Term MCS to improve end-organ function and to make a patient eligible for HT
BTT (Bridge to Transplant)	Long-Term MCS to keep alive a patient at high risk of death until an organ donor becomes available
BTR (Bridge to Recovery)	Long-Term-Short-Term MCS to keep a patient alive until the cardiac function recovers sufficiently to remove MCS
DT (Destination Therapy)	Long-Term MCS in those patients with end-stage heart failure ineligible for HT

## Data Availability

Data are contained within the article.

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
