# Peer review of "Outcome Through the Years of Left-Ventricular Assist Devices Therapy for End-Stage Heart Failure: A Review"

_jcm, 2024, doi:10.3390/jcm13216622_

Round 1
Reviewer 1 Report
Comments and Suggestions for Authors Complete the tex: - How long is survival with LVAD to Tx heart? - What are the new therapeutic methods (conservative) before LVAD? - What are the recommendations for patients with combined End-Stage Heart failure and End-Stage Kidney Disease? - Is simultaneous Tx recommended and to whom? - What is the recommended dose and which immunosuppressive therapy? - In GIT bleeding, is the HAS BLED score or the CHA2 DS2 VASc score more important - expand that section? Comments on the Quality of English Languagegood
Author Response
- How long is survival with LVAD to Tx heart? Thank you for the advice, I added some lines on different survival rates in the "outcome of patients with LVAD" paragraph (5).
- What are the new therapeutic methods (conservative) before LVAD? - What are the recommendations for patients with combined End-Stage Heart failure and End-Stage Kidney Disease? - Is simultaneous Tx recommended and to whom? - What is the recommended dose and which immunosuppressive therapy? - In GIT bleeding, is the HAS BLED score or the CHA2 DS2 VASc score more important - expand that section? Thank you for your advice, but I think these topics have a focus on heart transplantation, not on patients with LVADs.
Reviewer 2 Report
Comments and Suggestions for Authors
The manuscript offers a comprehensive review of Left Ventricular Assist Devices (LVADs) and their role in the treatment of end-stage heart failure. It traces the technological evolution of LVADs and presents a well-rounded discussion on the long-term outcomes and complications associated with their use. While it provides a thorough historical overview, the paper would benefit from a more balanced discussion on the latest advancements and practical clinical management.
Please include a dedicated section on recent and emerging technologies in LVADs. This would provide a forward-looking perspective on how these new innovations could impact future patient outcomes and clinical practice.
There is limited discussion of LVAD therapy in certain subgroups, such as elderly patients or those with significant comorbidities. Consider adding a section on the role of LVADs in these populations, discussing specific challenges, outcomes, and how management strategies might differ. It could be useful to refer to recent data from registries like INTERMACS that address these topics.
A figure summarizing the complications associated with LVADs would add value to the manuscript.
Moreover, the authors suggestions to minimize such complications would be interesting.
A minor revision of English is required.
Comments on the Quality of English LanguageA minor revision of English is required.
Author Response
The manuscript offers a comprehensive review of Left Ventricular Assist Devices (LVADs) and their role in the treatment of end-stage heart failure. It traces the technological evolution of LVADs and presents a well-rounded discussion on the long-term outcomes and complications associated with their use. While it provides a thorough historical overview, the paper would benefit from a more balanced discussion on the latest advancements and practical clinical management.
Please include a dedicated section on recent and emerging technologies in LVADs. This would provide a forward-looking perspective on how these new innovations could impact future patient outcomes and clinical practice.
Thank you for this advice, I added a few lines on some new techologies being studied (last paragraph).
There is limited discussion of LVAD therapy in certain subgroups, such as elderly patients or those with significant comorbidities. Consider adding a section on the role of LVADs in these populations, discussing specific challenges, outcomes, and how management strategies might differ. It could be useful to refer to recent data from registries like INTERMACS that address these topics.
Thank you for the advice, I added a section in the "Outcome of patients with LVAD" paragraph. Datas are from recent studies and from the INTERMACS registry.
A figure summarizing the complications associated with LVADs would add value to the manuscript.
I am working on a figure, but it is not still ready.
Moreover, the authors suggestions to minimize such complications would be interesting.
A minor revision of English is required.
A revision of english was done.
Round 2
Reviewer 1 Report
Comments and Suggestions for Authors
No
Author Response
Thank you for your revision.
Reviewer 2 Report
Comments and Suggestions for Authors
Suggestion: A figure summarizing the complications associated with LVADs would add value to the manuscript.
Authors response: I am working on a figure, but it is not still ready.
Please resubmit when the figure will be ready.
Author Response
Thank you for your suggestion, I submitted the figure.